# A Novel Megakaryocyte Subpopulation Poised to Exert the Function of HSC Niche as Possible Driver of Myelofibrosis

**DOI:** 10.3390/cells10123302

**Published:** 2021-11-25

**Authors:** Anna Rita Migliaccio

**Affiliations:** 1Department of Medicine and Surgery, Campus Bio-Medico University, 00128 Rome, Italy; amigliaccio@unicampus.it or amigliaccio.altius.org; 2Altius Institute for Biomedical Sciences, Seattle, WA 98121, USA

**Keywords:** myelofibrosis, megakaryocytes, thrombopoiesis, transforming growth factor beta (TGF-beta)

## Abstract

Careful morphological investigations, coupled with experimental hematology studies in animal models and in in vitro human cultures, have identified that platelets are released in the circulation by mature megakaryocytes generated by hematopoietic stem cells by giving rise to lineage-restricted progenitor cells and then to morphologically recognizable megakaryocyte precursors, which undergo a process of terminal maturation. Advances in single cell profilings are revolutionizing the process of megakaryocytopoiesis as we have known it up to now. They identify that, in addition to megakaryocytes responsible for producing platelets, hematopoietic stem cells may generate megakaryocytes, which exert either immune functions in the lung or niche functions in organs that undergo tissue repair. Furthermore, it has been discovered that, in addition to hematopoietic stem cells, during ontogeny, and possibly in adult life, megakaryocytes may be generated by a subclass of specialized endothelial precursors. These concepts shed new light on the etiology of myelofibrosis, the most severe of the Philadelphia negative myeloproliferative neoplasms, and possibly other disorders. This perspective will summarize these novel concepts in thrombopoiesis and discuss how they provide a framework to reconciliate some of the puzzling data published so far on the etiology of myelofibrosis and their implications for the therapy of this disease.

## 1. Introduction

It is long known that myelofibrosis, the end stage of the Philadelphia-negative myeloproliferative neoplasms (MPN) [1,2], is characterized by the presence of great numbers of morphologically immature megakaryocytes, in turn characterized by low ploidy levels and reduced presence of granuli and platelet-territories in their cytoplasm, in the bone marrow [3]. Insights in the mechanism(s) that retain megakaryocytes immature in this disease are provided by the observation that the malignant megakaryocytes contain low levels of GATA1 [4,5], the transcription factor necessary for their maturation [6]. The GATA1 content is thought to be maintained low in these cells by a ribosomopathy induced by the driver mutations, which selectively impairs binding of *GATA1* mRNA to the ribosomes, reducing its translation [5]. The causative role exerted by these immature megakaryocytes in the development of myelofibrosis was then confirmed by the observation that the hypomorphic *Gata1* mutation, which selectively decreases Gata1 transcription in megakaryocytes [7,8], induces myelofibrosis in mice [9]. Given that in myelofibrosis the immature megakaryocytes contain high levels of the pro-fibrotic growth factor TGF-β [10,11,12,13], these cells are thought to induce fibrosis by releasing TGF-β, which stimulates the cells of the bone marrow microenvironment to secrete and polymerize collagen [14,15]. These concepts have inspired ongoing clinical trials, which are testing the efficacy of treating myelofibrosis patients with drugs that either increase GATA1 content, restore megakaryocyte maturation [16], or inhibit TGF-β signaling [17].

This coherent and comfortable picture on the role exerted by immature megakaryocytes in the etiology of myelofibrosis has been recently challenged by observations obtained by single cell profiling and lineage tracking of primary megakaryocytes purified from different sources and by the identification of possibly alternative routes for their differentiation (discussed in detail below). These novel observations indicate that we probably still do not fully understand what “immature megakaryocytes” in the bone marrow of primary myelofibrosis are, from which cells they derive, and by which micro environmental cues they are regulated. As discussed below, this gap of knowledge mandates for novel investigation to clarify the role exerted by megakaryocytes abnormalities in the etiology of myelofibrosis.

## 2. Novel Insights into the Megakaryocyte Differentiation Pathway Have Identified a Novel Population of Cells Poised to Exert Niche Functions Which May Represent Drivers for Myelofibrosis

Several laboratories are currently using single cell profiling and lineage tracking of megakaryocytes purified for primary sources to generate a comprehensive model for the progression and routes of megakaryocyte maturation in vivo. These studies have identified that in addition to cells poised to generate platelets (Plt, Plt-poised megakaryocytes), the hematopoietic stem cells (HSC) in the bone marrow generate megakaryocytes that are poised to exert immune (immune-poised) function in the lung and niche (niche-poised) functions in organs in need of tissue repair [18,19,20,21] (Figure 1). The maturation of both Plt-poised and immune-poised megakaryocytes is intrinsically controlled by the balance between the transcription factors GATA1 (high expression) and GATA2 (low expression) [18,19] while that of niche poised cells is driven by low levels of *GATA1* and high levels of *GATA2*. By contrast, with Ptl- and immuno-poised cells, niche-poised megakaryocytes express high levels of genes involved in extracellular matrix formation and TGF-β response (Figure 1). These findings are consistent with the emerging hypothesis that megakaryocytes exert organ-specific functions [21,22,23] and mandate for a re-classification of megakaryocyte maturation. Unfortunately, morphological investigation alone is unable to discriminate between immature Plt-poised MK from mature immune-poised and niche-poised cells since all these cell types are diploid and all express CD42b and CD41. Sun et al. have recently suggested that immuno-poised megakaryocytes specifically express CD53 [24], while Liu et al. [25] suggested instead that these cells are uniquely characterized by CD148 and CD48 expression. No marker has been identified so to distinguish between niche-poised and Plt-poised megakaryocytes. Further studies are therefore required to identify markers that may discriminate between these three populations to be used for their prospective isolation and to detail their functions and the events that regulate their differentiation.

Since the morphology of immature megakaryocytes poised to generate Ptl and that of mature immune- and niche-poised megakaryocytes is the same (low level of ploidy and of granules and platelet-territories in their cytoplasm), morphological observations may not establish whether the ribosomopathy in myelofibrosis (and the *Gata1*^low^ mutation in mice) blocks the maturation of Ptl-poised megakaryocytes (as hypothesized up to now) or rather switches the maturation of the megakaryocytes in the bone marrow from that of Ptl-poised to that of niche-poised cells, resulting in fibrosis. The latter hypothesis is in contrast with data provided by the Verstovsek laboratory, indicating monocyte-derived fibrocytes as responsible for the fibrosis developed in myelofibrosis [26] but consistent with the intriguing data generated by the Balduini laboratory, indicating that, while megakaryocytes expressing collagens are very rare in the bone marrow from healthy individuals, they represent a great proportion of those present in the bone marrow of patients with myelofibrosis [27,28]. Whether the megakaryocytes observed by the Balduini laboratory in myelofibrosis are niche-poised cells containing low levels of GATA1 induced by the high levels of transforming growth factor-β (TGF-β) present in the microenvironment has not been established as yet. Since monocytes purified from the bone marrow of myelofibrosis patients may induce the formation of megakaryocytes and the development of myelofibrosis when transplanted in immunodeficient mice [29], the possibility exists that the population responsible for fibrosis identified by Verstovsek et al. and by Balduini et al. is one and the same, that is, niche poised-megakaryocytes. Further studies are necessary to test this hypothesis.

The current model of megakaryocytopoiesis indicates that megakaryocytes are generated from hematopoietic stem cells (HSC) and progressively mature under the control of intrinsic factors (i.e., the transcription factors *GATA1* and *GATA2*) and extrinsic factors (i.e., thrombopoietin (TPO) and stromal cell-derived factor 1 (SDF-1) [30]. Based on distinctive morphological markers, megakaryocyte precursors are divided into four classes of progressively more mature cells: the pro-megakaryoblast (Stage 0), the megakaryoblast (Stage I), the promegakaryocyte (Stage II), and finally, the mature megakaryocyte (Stage III), which is capable of releasing platelets (Plt) [31]. Recent single cell RNAsequencing (RNAseq) data and lineage tracking analyses indicate that Stage I megakaryocytes may generate, in addition to platelet-poised megakaryocytes, cells that are poised to exert either immune or niche functions [18,19,20,21]. Since the morphology of the megakaryocytes poised to exert the different functions is the same, morphological analyses alone may not determine the relative frequency of the three populations in various tissues. We suggest that, as hypothesized by the Balduini laboratory [27,28], the GATA1 hypomorphic immature megakaryocytes found in the bone marrow of patients with myelofibrosis are niche-poised cells and are directly responsible for fibrosis.

## 3. Endothelial Cells (EC), a New Source of Megakaryocyte Precursors Which May Be Responsible to Generate the Megakaryocytes That Drive Myelofibrosis

Myelofibrosis is the end-stage phase of the Philadelphia-negative MPNs, a continuum class of diseases driven by mutations in genes of the thrombopoietin axes (*MPL*, *JAK2*, or *Calreticulin*) occurring at the level of the HSC and that alter its proliferation [1]. Most of what is known of the mechanisms that guide progression of MPN to myelofibrosis has been obtained by studying models harboring the *JAK2*V617F mutations, under the assumption that similar mechanisms drive the disease also in patients harboring different mutations [1]. Two cell populations are thought to be responsible for driving fibrosis in the bone marrow: *JAK2*V617F-megakaryocytes generated by the malignant HSC, which remain immature and present a pro-fibrotic phenotype very similar to that expressed by the niche-forming megakaryocytes (Figure 1 and [18,32]), and *JAK2*V617F-ECs, which do not derive from HSC, found in the blood vessels from the liver and spleen (but also in the marrow) of MPN patients [33,34]. Data in animal models supporting the pathobiological role of both cell populations in the etiology of the disease exist. In fact, lineage-restricted expression of *JAK2*V617F in mice in either the megakaryocytes or the ECs has been shown to be sufficient to promote the development of the MPN phenotype even if the hematopoietic stem cells of these mice are normal [35,36,37]. However, since the EC do not derive from the HSC and myelofibrosis is a clonal defect of the HSC [38], the presence of the *JAK2*V617F mutation in EC and their role in the pathogenesis of the disease are puzzling.

Wang et al. [18] greatly advanced our understanding of the ontogenesis and cytogenesis of megakaryocytopoiesis by identifying a subpopulation of thrombospondin1-positive ECs derived from human embryonic stem cells (hESC) that, following induction, differentiate into megakaryocytes in vitro. Lineage-tracing markers in mice had previously identified that definitive HSC arise in the aorta–gonad–mesonephric region of the embryos from the hemogenic endothelium which gives rise, by asymmetric division, to resident ECs and HSCs, which are released into the blood and subsequently colonize the liver [39]. The Peault laboratory then described the presence of definitive HSCs in the aorta–gonad–mesonephric region of human embryos that were capable of colonizing adult xenografts [40] and reported that definitive HSCs were derived from hemogenic endothelium generated in culture from hESC that resemble those observed in mouse embryos. The fact that the timing at which thrombospondin1-positive EC are detected in hESC culture (day 5) is similar to when hESC generate the hemogenic endothelium described by Peault [41] suggests that the hemogenic endothelium gives rise at the same time to HSC and to ECs posed to generate niche-poised megakaryocytes, motile cells that may reach the liver to support its colonization by the HSC (Figure 2). Although thrombospondin-positive EC generate in vitro mostly immune- and Ptl-poised megakaryocytes, it should be mentioned that the microenvironment of the embryo discourage the differentiation of immune cells. Therefore, we hypothesize that in vivo megakaryocyte-biased EC are guided toward the niche-poised lineage rather than the immune-poised one. This hypothesis, which needs to be tested by dedicated experiments, provides a coherent model for the generation of the HSC and of their supporting niche from a common precursor during ontogenesis.

The hypothesis that thrombospondin1-positive ECs capable of generating HSC-supportive megakaryocytes may persist in adults clarifies the current puzzling data dealing with the cell culprit (HSC or endothelial cells) for the initiation of the Philadelphia-negative MPN. The MPNs are a group of blood cancers characterized by hyperproliferation of hematopoietic cells of one or more lineages in extramedullary organs [42]. A clue that genetic alterations that promote the development of MPNs may occur in precursors of the HSC before birth is provided by the observation that infants with Downs Syndrome who accumulate during their fetal life deletions in the GATA1 gene supporting expression only of a hypomorphic isoform of the GATA1 protein are prone to develop a transient MPN, which eventually progresses to acute megakaryocytic leukemia [31,43]. The data by Wang et al. [18] suggest that this sequence of events, as first hypothesized by the Orkin laboratory [44], is due to low levels of functional GATA1, which favor differentiation of niche-supporting megakaryocytes in these children (Figure 2). The extensive HSC proliferation supported by these megakaryocytes after birth would then favor accumulation of additional HSC mutations and progression to acute leukemia. These findings are in synchrony with the recent intriguing report of Williams et al., indicating that the *JAK2*V617F mutation may be first acquired during fetal development, while the actual MPN phenotype only becomes apparent during adulthood [45]. The confluence of these observations suggests that some hematological malignancies may arise during fetal development and that they are induced by mutations in the hemogenic endothelium, which may represent the embryonic equivalent of the hemangioblasts found in adult tissues, which generates both the HSC and their supporting niche. This hypothesis would explain why the *JAK2*V617F mutation is detected both in hematopoietic and endothelial cells. Whether the *JAK2*V617F mutation first arises in the hemogenic endothelium, which generates both the HSC and the thrombospondin1-positive EC capable of generating megakaryocyte-primed EC, and whether these thrombospondin1-positive EC persist in myelofibrosis, is worthy of further investigation.

These photos depict the great hyperproliferation of HSC occurring in extramedullary sites both during development and in MPN. The paper by Wang et al. [18] provides data inferring that niche-forming megakaryocytes, possibly generated by thrombospondin1-positive EC derived from the hemogenic endothelium, that may represent the embryonic equivalent of the hemangioblasts found in adult tissues, may be responsible for the hyperproliferation of the HSC observed in the liver during embryonic development and, driven by *JAK2*V617F, in the extramedullary sites in MPN.

## 4. Conclusions

Emerging concepts on megakaryocytopoiesis are revolutionizing what we knew on this process and on what are its abnormalities, which are involved in the etiology of myelofibrosis, the most severe of the Philadelphia-negative MPN. In addition, to provide a framework to re-conciliate contradictory results currently published on this disease, these emerging concepts suggest new avenues of investigation that hold great promise to ameliorate the prognosis and the therapy for this unmet clinical need.

## Figures and Tables

**Figure 1 cells-10-03302-f001:**
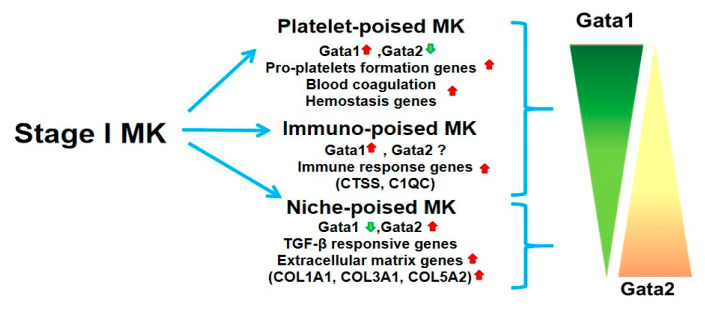
Novel model of megakaryocyte maturation based on single cell profiling of prospectively isolated embryonic and adult megakaryocytes. The question marker indicates that the level of GATA2 in these cells are unknown.

**Figure 2 cells-10-03302-f002:**
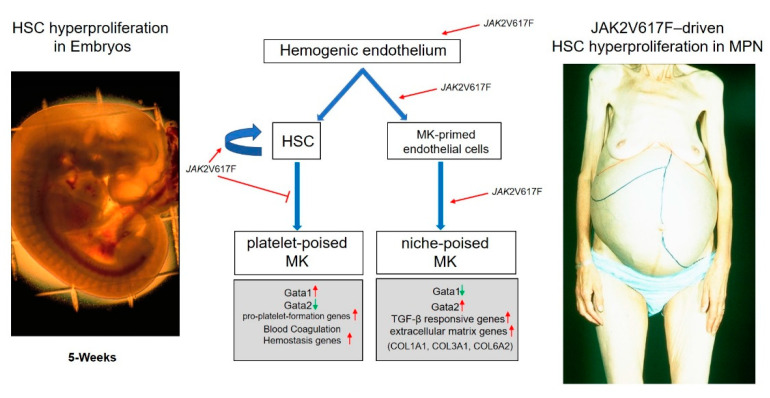
Photographs of one human embryo with a large liver rudiment at 5 weeks post-conceptus (**left**) and of the enlarged abdomen, a reflection of the greatly enlarged spleen and liver due to massive extramedullary hematopoiesis, of one patient with MPN (**right**).

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
