# Peer review of "A Novel Megakaryocyte Subpopulation Poised to Exert the Function of HSC Niche as Possible Driver of Myelofibrosis"

_cells, 2021, doi:10.3390/cells10123302_

Round 1

Reviewer 1 Report

In this prospective, Dr. Migliaccio expertly discusses contemporary controversies and future directions in MF pathogenesis driven by MK subpopulations. Overall, the monograph is well written and explains complicated concepts. Most importantly, the author lays out a framework for future investigations to advance the field. 

Major comment: 

  • The EC precursor discussion is excellent, however the author does not discuss an alternative explanation - the concept of a hemangioblast which can explain the presence of JAK2 in both the EC and the HSC. A brief discussion of this concept, including its faults, would strengthen this section

Minor comments:

  • Page 1 line 19: Should be “and possibly other disorders”
  • Page 2 line 44: Should be “comfortable”
  • Page 2 line 79: Should be “up”
  • Page 2 line 83: Should be “fibrocytes”

Author Response

Thank you very much for your kind words of appreciation. It means a lot to me. The oversights identified in your review have been all addressed. The modifications are indicated in red fonts in the revised manuscript. 

Reviewer 2 Report

In this Perspective, Anna Rita Migliaccio offers an interesting proposal about the possible role of a recently identified specific megakaryocyte (MK) subset in the pathogenesis of myelofibrosis (MF). 

The Author comments on how new single-cell technologies over the last years have enlarged and added complexity to the landscape of human megakaryopoiesis.

In particular, in the first part of the manuscript her main focus is on a recent study from Sun S et al. identifying 3 different MK subpopulations: immune-, niche-supporting and platelet-generating MK, called by AR Migliaccio as "immune-, niche- and platelet-poised" MK, respectively. She highlights how the niche-poised MK resemble the low-GATA1 MK found in MF: in fact, their maturation is driven by low levels of GATA1 and they express high levels of genes involved in extracellular matrix formation and TGF-β response.

In the second part of her work, the Author discusses about the recent evidence from Wang et al. of a MK-biased thrombospondin-1+ endothelial cell (EC) population derived from human embryonic stem cell. The latter cells were previously shown to produce both haematopoietic stem cells (HSC) and EC. As the MK-biased EC population is able to differentiate to MK, the Author postulates that these cells could be the precursors of niche-poised MK, which then migrate to extramedullary sites and support the growth of HSC in other organs in MF patients. This hypothesis would explain the abnormal extramedullary haemopoiesis in MF and the finding of JAK2V617F-positive EC in MF patients, the latter finding deriving from the driver mutation arising in a common embryonic HSC/EC precursor.

I am in favour of the publication of this Perspective as it offers a good hint for future research on the mechanism of MK contribution in MF, cleverly bringing together similarities between evidence on abnormal MK in MF and recent scientific discoveries on human megakaryopoiesis.

I have however some Major and Minor points to be addressed/clarified.

Major:

  1. Line 70: “morphological investigation alone is unable to discriminate between immature Plt-poised MK from immune-poised and niche-poised cells since all these cell types are diploid and all express CD42b and CD41”. Actually, in the work from Sun S et al. the immune-supporting MK subset is clearly distinguished by the other two by the low-ploidy and the CD53-positivity. Can the Author clarify this point?
  2. Lines 114 and 145: in my opinion, in these passages the Author simplifies too much the concepts on MF pathogenesis. In line 114 AR Migliaccio states that “Two cell populations are thought to be responsible for driving MPN development: JAK2V617F-positive megakaryocytes….and JAK2V617F-ECs found in the blood vessels from the liver and spleen of MPN patients”. First of all, no mention is done about the CALR-, MPL-mutated and triple-negative MF, which can’t be excluded from the discussion. Secondly the two above mentioned cells are not the only players in the MPN (and MF) pathogenesis: HSC themselves, monocytes and mesenchymal stromal cells are other important contributors to MF development. Similarly, in line 145, the Author seems to suggest that currently there are puzzling data about which cell gives rise to MF, the HSC or the EC. Actually, extensive literature points toward the multipotent HSC as the initiating cell in MPN, including MF. Data on EC are still limited. Can the Author re-write more appropriately these two passages?
  3. Lines 138-140: “the hemogenic endothelium gives rise at the same time to HSC and to EC posed to generate niche-forming megakaryocytes, motile cells that may reach the liver to support its colonization by the HSC”. Actually, in the work from Wang et al., the MK subset originating from MK-biased EC is the ‘‘immune’’ MK subpopulation and not the “niche-supporting” one. Can the Author motivate better how the thrombospondin-1-positive EC cells could give rise to the “niche-poised” MK subset?

Minor:

  1. Line 44-52: the Author states that recent discoveries challenge the classical theory by which the low-GATA1 MK are crucial players in the aetiology of MF. Can the Authors explain better why? From what presented in this Perspective it is seems that the recent findings on megakaryopoiesis help to redefine what we know so far about the role of MK in the pathogenesis of MF, without going against it. Also, can the Author provide a scientific definition of “immature MK” in MF?
  2. Line 71 and 76: the Author writes “immature megakaryocytes poised to generate Ptl” referring to the platelet-generating MK identified by Sun S et al. Actually, Sun S and co-workers identified 3 subpopulation of mature Can AR Migliaccio clarify why she defines this MK subset as “immature”?
  3. The Author should adjust the paragraph numbering (1,2,3,4 instead of 1,2,4,4)
  4. Legend to Figure 1, line 97: did the Author mean “megakaryopoiesis” instead of “thrombopoiesis”?

Author Response

We thank this reviewer for the nice summary of our perspective and hot her/his words of appreciation. The comments made by the reviewers are excellent and have been carefully addressed to improve clarity and precisions. The changes are summarized below and indicated in red fonts in the revised manuscript.

Point-by-point replay to the reviewer' comment. 
